# Traffic-Related Air Pollution and Breast Cancer Risk: A Systematic Review and Meta-Analysis of Observational Studies

**DOI:** 10.3390/cancers15030927

**Published:** 2023-02-01

**Authors:** Delphine Praud, Floriane Deygas, Amina Amadou, Maryline Bouilly, Federica Turati, Francesca Bravi, Tingting Xu, Lény Grassot, Thomas Coudon, Béatrice Fervers

**Affiliations:** 1Prevention Cancer Environment Department, Centre Léon Bérard, 28 rue Laënnec, 69008 Lyon, France; 2Inserm, U1296 Unit, “Radiation: Defense, Health and Environment”, Centre Léon Bérard, 28 rue Laënnec, 69008 Lyon, France; 3Department of Clinical Sciences and Community Health, University of Milan, Via A. Vanzetti 5, 20133 Milan, Italy

**Keywords:** air pollution, traffic, nitrogen dioxide, epidemiology, breast cancer, meta-analysis

## Abstract

**Simple Summary:**

We reviewed 21 epidemiological studies on breast cancer risk and exposure to traffic-related air pollution. The exposure assessment methodologies were heterogeneous. There was variability across studies on temporal concordance between the exposure periods relevant to breast cancer and the time period of the exposure assessment. There was little evidence of an association between traffic-related air pollution estimated with proxies and breast cancer risk. The random-effects meta-analysis of 13 studies on NO_2_ and NO_x_ exposure showed an increased risk of breast cancer with NO_2_ exposure.

**Abstract:**

Current evidence of an association of breast cancer (BC) risk with air pollution exposure, in particular from traffic exhaust, remains inconclusive, and the exposure assessment methodologies are heterogeneous. This study aimed to conduct a systematic review and meta-analysis on the association between traffic-related air pollution (TRAP) and BC incidence (PROSPERO CRD42021286774). We systematically reviewed observational studies assessing exposure to TRAP and BC risk published until June 2022, available on Medline/PubMed and Web of Science databases. Studies using models for assessing exposure to traffic-related air pollutants or using exposure proxies (including traffic density, distance to road, etc.) were eligible for inclusion. A random-effects meta-analysis of studies investigating the association between NO_2_/NO_x_ exposure and BC risk was conducted. Overall, 21 studies meeting the inclusion criteria were included (seven case–control, one nested case–control, 13 cohort studies); 13 studies (five case–control, eight cohort) provided data for inclusion in the meta-analyses. Individual studies provided little evidence of an association between TRAP and BC risk; exposure assessment methods and time periods of traffic emissions were different. The meta-estimate on NO_2_ exposure indicated a positive association (pooled relative risk per 10 µg/m^3^ of NO_2_: 1.015; 95% confidence interval, CI: 1.003; 1.028). No association between NO_x_ exposure and BC was found (three studies). Although there was limited evidence of an association for TRAP estimated with proxies, the meta-analysis showed a significant association between NO_2_ exposure, a common TRAP pollutant marker, and BC risk, yet with a small effect size. Our findings provide additional support for air pollution carcinogenicity.

## 1. Introduction

Among the wide range of air pollution sources, traffic is one of the main ones in urban areas. Traffic-related air pollution (TRAP) is composed of several gaseous pollutants and particulate matters (PM) [1]. The published literature on the human health effects of exposures to these pollutants is increasing [2,3]; in particular, several epidemiological studies have reported that long-term exposure to TRAP is associated with the development of certain cancers. The International Agency for Research on Cancer (IARC) has classified diesel exhaust and PM, as well as outdoor air pollution overall as carcinogenic to humans [4,5], with sufficient evidence for a causal link with lung cancer [6,7]. Because of the long latency period between exposure and cancer occurrence, the study of the carcinogenic effects of traffic-based pollutants is difficult and requires large and long-term observational studies.

As motor-vehicle traffic is generally the main emission source of nitrogen dioxide (NO_2_) or NO_x_ [8,9], exposure to NO_2_ can be considered as a surrogate of exposure to TRAP. This assumption has been greatly supported by the recent fall of NO_2_ levels during coronavirus lockdowns, a period during which travel and commutes were drastically reduced [10]. NO_x_ and NO_2_ are less considered for their carcinogenicity than for their specificity as markers of TRAP. It is worth noting that NO_2_ is strongly correlated to other air pollutants with plausible biological mechanisms for breast cancer, such as ultrafine particles (UFPs) or polycyclic aromatic hydrocarbons (PAHs) [11,12,13,14].

Current epidemiological evidence on the association of air pollution exposure and, more specifically, TRAP with breast cancer risk remains inconclusive [15,16]. The IARC working group concluded in 2013 that the evidence of the carcinogenicity of air pollution for breast cancer was based on a relatively small number of informative studies and that the observed associations were inconsistent [5]. A recent meta-analysis reported an association between increasing exposure of 10 µg/m^3^ of NO_2_ and an increased risk of 3% of breast cancer, but no significant association with PM (PM_2.5_ and PM_10_) exposure was found [17]. However, a positive association has been shown for breast cancer mortality for an increasing of 10 µg/m^3^ of PM exposure [18]. Since some inhaled toxic substances have been found in breast fluid [19], it is suspected that air pollutants can reach breast tissue. Moreover, experimental evidence from mechanistic studies concluded that an association between air pollution and breast cancer risk is biologically plausible. In particular, benzene, a non-methane volatile organic compound (NMVOC) present in traffic exhaust, has been linked to the development of mammary tumors in mice [20]. Moreover, PAHs originating from traffic emissions may cause breast cancer through DNA damage [13], aberrant DNA methylation [14], and estrogenic and antiestrogenic activities [11].

In the present study, we conducted a systematic review of the published literature on breast cancer risk associated with TRAP exposure, including studies using qualitative or semi-qualitative estimates (e.g., distance from roadways or traffic volume on nearby roadways), as well as studies assessing the main pollutants produced from automobile vehicle emissions (i.e., CO, NO_2_, NO_x_, NMCOV, PM_10_, and PM_2.5_). Additionally, we performed a meta-analysis of a sufficiently homogenous group of studies on NO_2_ and NO_x_, a common marker for TRAP. Among women, we evaluated the effect of the highest exposure compared to the lowest exposure on breast cancer incidence or the incremental effect of 10 µg/m^3^ exposure to NO_2_ or NO_x_ on breast cancer incidence.

## 2. Materials and Methods

The scope of this systematic review was framed according to the PECO strategy (i.e., population; exposure; comparison; outcome) [21]: we included studies on women (population), analyzing exposure to TRAP (exposure) compared to less-exposed women (comparator) on the risk of breast cancer (outcome). This study was registered with PROSPERO (CRD42021286774).

### 2.1. Search Strategy

We conducted the literature review following the Preferred Reporting Items for Systematic Reviews and Meta-analysis statement (PRISMA, http://www.prisma-statement.org accessed on June 2022) (Supplementary Material S1). We identified informative studies published until June 2022 through a systematic search on the Medline/PubMed and Web of Science databases. The core search consisted of key words related to traffic air pollution and breast cancer; the search algorithm used is provided in the Appendix A. Human-based epidemiological studies written in English, French, German, and Italian were considered. The electronic search was supplemented by hand searching of references from previous reviews on the issue, as well as the reference lists of the identified publications [22,23,24,25]. 

### 2.2. Study Selection

Two reviewers (D.P. and F.D.) independently selected eligible studies by screening titles, abstracts, and, when appropriate, full-texts of articles. Any discrepancy was discussed and resolved by consensus. Studies were eligible if they were an original observational study (case–control or cohort) and responded to the defined PECO strategy. We included studies using a proxy of exposure based on measures of traffic density, distance to or length of road, as well as those using exposure modelling approaches (e.g., dispersion modelling, land use regression models, and kriging models) for assessing exposure to the main pollutants produced from automobile vehicle emissions (i.e., CO, NO_2_, NO_x_, NMCOV, PM_10_, PM_2.5_, and black carbon). Studies further had to report measures of association (i.e., odds ratios, ORs; relative risks, RRs; or hazards ratios, HRs) and the corresponding 95% confidence intervals (95% CIs). Studies on genes or protein expression, polymorphisms, gene–environment interactions, survival, cancer recurrence, mortality, and occupational or industrial exposures, as well as ecological studies were not included. Studies on male breast cancers were not considered.

For the quantitative analysis, when there were multiple publications from the same cohort study and various follow-up periods, we chose studies using the longest follow-up time. For the ESCAPE study (a consortium of European cohorts) [26] including 15 studies, we used the pooled estimates of the consortium rather than the individual study estimates.

### 2.3. Data Extraction

For each selected publication, details on the study design, country, number of enrolled subjects, sources of case recruitment, number and type of breast cancer (invasive or in situ), period of enrolment (for case–control studies) or follow-up (for cohort studies), menopausal status at baseline, exposure variable(s), unit of exposure, timing and methods for exposure assessment, subjects’ exposure average, confounding factors for the adjustment of statistical models and results (overall and, when available, according to the methods of exposure assessment, menopausal status, and hormone receptor status of breast cancer) were extracted. When several statistical multivariable models were reported, we selected the one with the largest number of covariates, while paying attention to the homogeneity of the models across studies. When necessary, we contacted the corresponding authors to confirm data details (i.e., precision on adjustment variables, increase of exposure unit, and details on the population included).

### 2.4. Quality Assessment

The quality of each selected study was assessed by the Newcastle-Ottawa scale (NOS), a semi-quantitative assessment of study quality [27]. This instrument is based on eight items grouped into three domains: selection (4 items), comparability (1 item), and outcome (for cohort studies; 3 items) or exposure (for case–control studies; 3 items). Using a star rating system, a maximum of one star can be allocated for each item, except the comparability item, which can be assigned up to two stars, the highest-quality score a paper can obtain being 9. Two epidemiologists (D.P. and M.B.) independently assessed the study quality; any disagreement was resolved through input from a third researcher (B.F.) to reach consensus. We considered studies using land use regression and kriging models for exposure assessment as the highest exposure assessment quality, since these models are fit on ambient air measurement data. For cohort studies, a minimum of 10 years was considered as a sufficient follow-up time. As we found no universally accepted criterion for the definition of good quality based on the NOS score, we considered a score of 8–9 out of 9 as high quality, 6–7 as good quality, 4–5 as moderate quality, and ≤3 as low quality.

### 2.5. Statistical Analyses

For performing a meta-analysis, exposures assessed in at least three studies and estimated using homogeneous metrics were considered. This was the case for studies on NO_2_ and NO_x_ exposure. To allow a comparison of the NO_2_ and NO_x_ exposure effects among the different studies, we harmonized the measures of association of each study to express an impact of a 10 µg/m^3^ increase of exposure. Exposure estimates that were expressed in parts per billion (ppb) were converted to µg/m^3^ using a factor of 1.88 µg/m^3^ per 1 ppb, assuming ambient pressure to be equal to 1 atmosphere and a temperature of 25 °C [7,28]. If different models were used to assess NO_2_ or NO_x_ exposure in the same study, we chose estimates from the finest model. For one study, associations were assessed using NO_2_ concentration measures at different time points: at enrollment, 10 years before enrollment, 10 years after enrollment, and using the NO_2_ mean concentration during the 10-year period before enrollment [29]. We chose to consider the association estimates of the latter, since it covers the longest exposure period before diagnosis.

Since NO_x_ includes both NO and NO_2_, we conducted meta-analyses on NO_2_ and NO_x_ separately. We derived summary estimates of the RR for an increase of 10 µg/m^3^ exposure to NO_2_ or NO_x_ using random-effects models, which account for the heterogeneity among the RR estimates [30]. Heterogeneity was evaluated through the χ^2^ test and quantified through the I^2^ statistic, which represents the proportion of total variation attributable to between-study variance [31,32]. Forest plots were used to provide a visual representation of study-specific RRs and summary RRs, with their 95% confidence interval (CI). Publication bias was investigated visually through funnel plots [32] and Egger’s test [33]. Statistical significance was considered at *p*-values < 0.05, two-sided. 

We conducted subgroup analyses for NO_2_ according to: study design (case–control, cohort), geographical area (North America, Europe), menopausal status at baseline (postmenopausal, premenopausal), exposure assessment method (LUR, other), address used for exposure assessment (at baseline, residential history), and hormone receptor status of breast cancer (estrogen receptors (ERs) and progesterone receptors (PRs)). For analyses in strata of menopausal status, as a sensitivity analysis, we further included, in an additional analysis, two studies that did not clearly perform analyses by menopausal status, but included women at the age of menopause. These were the study by Datzmann et al., where 90% of the cases were ≥50 years old at inclusion [34], and the study by Cheng et al., where 87% of subjects were ≥50 years old and 90% were postmenopausal at inclusion [35].

For NO_2_, the meta-estimate was re-calculated by excluding each study in turn in sensitivity analyses. We also carried out a cumulative meta-analysis on the basis of the year of publication to consider possible changes over time in the association with breast cancer risk. Furthermore, we calculated the meta-estimate considering individually all studies included in the ESCAPE study. Finally, we considered only high-quality studies (NOS score ≥ 8).

All statistical analyses were performed using R software Version 3.6.1.

## 3. Results

### 3.1. Studies Selected

Figure 1 shows our search and selection process for the review and the meta-analysis. From a total of 925 citations, 758 were excluded after title and abstract screening. After examining 58 full-text articles, 19 met the inclusion criteria for this review. Four additional articles were included after searching reference lists of previous reviews and relevant publications. Among the 23 articles considered, we did not include in the qualitative synthesis the study by Niehoff et al. analyzing only the subgroup of postmenopausal women [36], since the study by Mordukhovich et al. (based on the same study, considering both premenopausal and postmenopausal women) was already included [37]. Similarly, we did not include one cohort [38] already considered through a more recent article [39]. Overall, 21 studies meeting the inclusion criteria were included in this systematic review. 

For NO_2_ or NO_x_ exposure, the number of available studies using homogenous metrics and estimating the association with breast cancer risk was sufficient to conduct a meta-analysis. Overall, 15 studies provided association measures for NO_2_ or NO_x_. For the meta-analyses on NO_2_/NO_x_, one cohort study was not included [22] as it was already considered through the ESCAPE consortium [26]. We also excluded one article that did not present the cut-off values of NO_2_ exposure categories [24]. Finally, 13 studies were included in the meta-analysis.

### 3.2. Review of Studies on Traffic-Related Air Pollution Exposure

Studies assessing the effect of TRAP exposure on breast cancer risk are presented in Table 1. Thirteen cohort studies [22,23,24,26,34,35,39,40,41,42,43,44,45], including a pooled analysis of 15 cohorts [26], seven case–control studies [25,29,37,46,47,48,49], and one nested case–control study [50], published between 1996 and 2022, were identified. Eight were conducted in the United States [35,37,39,42,43,45,46,47], five in Canada [23,29,41,48,49], two in Denmark [22,40], two in France [25,50], one in Israel [44], one in Germany [34], one in Taiwan [24], and one was a pooled analysis of 15 cohorts from nine European countries [26].

Regarding the enrollment period, three studies started in the 1980s [41,42,46], nine in the 1990s [22,29,35,37,40,45,47,48,50], and eight in the 2000s [23,24,25,34,39,43,44,49]. Enrolment periods started in 1980 in the earliest study and in 2014 for the most-recent one. For the pooled analysis of 15 cohorts, the period of enrollment varied between 1992 and 2005 according to the studies [26].

All cases were invasive breast cancers, except in seven studies that included also in situ cases [22,24,25,37,39,43,45,47]. Three studies were conducted only on postmenopausal women [26,29,49], and eleven studies performed subgroup analyses according to menopausal status [25,37,39,41,42,43,45,46,47,48,50]. The selection of cases was based on cancer registries in eight studies [23,26,35,40,41,44,45,48], hospital and physician list in six studies [25,29,37,46,47,49], self-administered questionnaires in six studies [22,26,39,42,45,50], and health insurance databases in two studies [24,34]. 

The metrics used to assess exposure to TRAP varied widely across studies (Table 1). Seven studies used spatial surrogate measures to assess TRAP exposure at the subjects’ residence. One study estimated traffic density by the number of vehicle-miles divided by the number of miles of highway [46], others by the distance from subject’s residence to the closest major road [22,42,48] or by the number of vehicles on the nearest road [22,26], and another by several characteristics of the main road and of the nearest cross-street [43]. For the distance from the subject’s residence to the closest major road, Hart et al. used residential proximity (0–49 m, 50–199 m, and ≥200 m) coupled with the type of road categories [42], Hystad et al. used the duration of residence at three categories of distance and for types of road (highways and major roads) [48], and Raaschou-Nielsen et al. used a binary distance variable (major street within 50 m, yes/no) [22]. Given the heterogeneity of the metrics used in these studies, performing a meta-analysis was not suitable. 

Two studies used dispersion models to estimate exposure to benzo[a]pyrene (BaP) or PAHs due to traffic releases [37,47]. 

Eleven studies analyzed NO_2_ exposure [23,24,25,29,34,39,40,41,45,48,49,50], one on NO_x_ [44] and two on both NO_2_ and NO_x_ [26,35]. Exposure assessment of these reports were based on LUR models, except for three studies that used kriging and dispersion models [25,39,40] and one study using measures from monitoring stations [24]. Moreover, some studies used several methods [35,48,50]. 

No articles on PM exposure were included because they did not focus on traffic emission and might have taken into account other major sources of PM exposure including residential heating, producing industry, or agriculture [51].

The results from the NOS assessment are presented in Table 1. Of the 21 articles included in the present review, more than 85 % were assessed as being of high quality (8 articles; NOS score ≥ 8) or good quality (10 articles; NOS score 6–7); three articles were of moderate quality (NOS score 4–5); no study was of low quality. Overall, no study was excluded based on quality, to further review the TRAP exposure assessment methods and the relationship between TRAP and breast cancer.

Figure 2 synthetizes the exposure assessment details on the addresses used, timing of exposure, timing of exposure assessment, and range of mean pollutant concentrations. Overall, eleven studies used the subjects’ residential history [23,24,25,35,37,40,42,46,47,48,50] and nine studies the baseline address [22,26,34,39,41,44,45,49,52] for exposure assessment. More specifically, regarding the timing of exposure, studies estimated exposures at baseline [22,26], during the follow-up until the index date [24,25,35,40,48,50,53], and during a period (7–20 years) prior to diagnosis or the index year [23,37,47,48]. One study was based on the longest residence before the age of 14 years [43]. Additionally, Nie et al. also focused on specific time windows, i.e., at menarche and at first child birth [47] (Table 1, Figure 2).

Seven studies estimated exposure at one point in time: before inclusion [34,43,49], at inclusion [22,26], at the midpoint of follow-up [39], or after the end of follow-up [41]. In one study, exposure was estimated at three points in time; before, after, and at inclusion [29], while another study assessed exposure 7 years before inclusion and during follow-up [23]. Finally, one study assessed exposure during follow-up, but using the address at inclusion [44,45] (Figure 2).

Except for the ESCAPE study, covering a wide range of NO_2_ exposure means across studies of the consortium ranging from 5.4 µg/m^3^ to 53.0 µg/m^3^, average NO_2_ exposure estimates across studies varied from 19.17 µg/m^3^ (for the study by White et al. assessing NO_2_ exposure in the U.S. in 2006 [39]) to 37.79 µg/m^3^ (for the study by Crouse et al. assessing NO_2_ exposure in Canada in 1985). Of note, exposure assessment for 2006, for the latter study, was 21.24 µg/m^3^ [29] (Figure 2).

Regarding the results of the association analyses, studies using spatial surrogate variables of TRAP exposure did not find significant associations with breast cancer risk, although one study reported a borderline association (HR = 1.4, 95% CI: 1.0; 1.9) between breast cancer risk and the proximity, during childhood, to a road with characteristics of high exposure to traffic-related pollutants (i.e., close proximity, presence of median/barrier, multiple lanes, and heavy traffic) [43] (Table 1). The study analyzing exposure to BaP showed a positive association between exposure at first child birth and breast cancer among postmenopausal women, with an OR of 2.58 (95% CI: 1.15; 5.83) for the highest versus the lowest quartile of exposure [47]. Regarding NO_2_ and NO_x_, all estimates were above 1 (the value of the null effect) with narrow confidence intervals. Increased risks were reported in one study for NO_2_ (RR = 1.07, 95% CI: 1.03; 1.12 per increase of 10 μg/m^3^) [34] and one study for NO_x_ (HR = 1.43, 95% CI: 1.12; 1.83) for a 10 ppb increase) [44] (Table 1). 

The heterogeneity of the exposure assessment methods did not allow performing an overall meta-analysis of TRAP exposure and breast cancer risk, except for studies investigating NO_2_/NO_x_ exposure.

**Figure 2 cancers-15-00927-f002:**
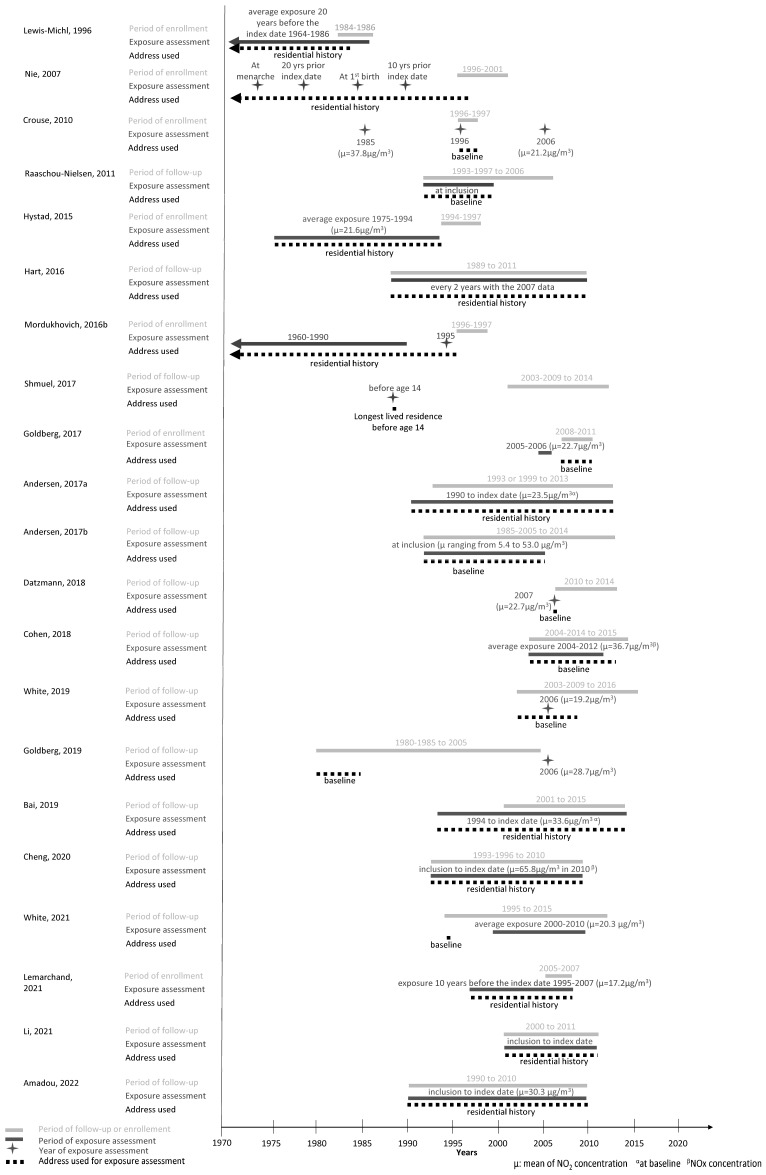
Summary of the temporal concordance between the period of enrollment/follow-up, exposure assessment, and address used for exposure assessment for each study included in the systematic review on breast cancer risk and exposure to traffic-related pollutants [22,23,24,25,26,29,34,35,37,39,40,41,42,43,45,46,47,48,49,50].

### 3.3. Meta-Analysis on NO_2_/NO_x_ Exposure

Figure 3 shows study-specific and pooled RRs, with their 95% CIs, of invasive breast cancer risk for an increase of 10 µg/m^3^ of NO_2_ exposure. The overall RR for the association between invasive breast cancer and 10 µg/m^3^ increase in NO_2_ was 1.015 (95% CI: 1.003; 1.028, *p* = 0.02), and heterogeneity across the thirteen studies was low and not significant (I^2^ = 17.0, *p* for heterogeneity = 0.27). There was no significant publication bias (*p* for Egger’s test = 0.06). The pooled estimate was 1.011 (CI 0.999, 1.024, I^2^ = 20.0, *p* for heterogeneity = 0.27) for the eight cohort studies, and 1.048 (CI 1.009, 1.088, I^2^ = 0.0, *p* for heterogeneity = 0.73) for the five case–control studies.

Table 2 presents the pooled RRs and 95% CIs according to the exposure assessment method, study geographical area, menopausal status, and hormonal receptor status. In the subgroup analyses by the exposure assessment method, based on ten studies using LUR models, the pooled RR was 1.016 (95% CI: 1.002; 1.030, I^2^ = 26.5, *p* for heterogeneity = 0.20), while for other methods (kriging in two studies, dispersion modelling in two studies, and chemistry transport models in two studies), the pooled RR was 1.037 (95% CI: 1.006; 1.069, I^2^ = 9.8, *p* for heterogeneity = 0.35).

Furthermore, subgroup analyses according to the address used for exposure assessment and the pooled RRs for studies estimating exposure at the baseline address were 1.018 (95% CI: 0.993; 1.043, seven studies, I^2^ = 38.7, *p* for heterogeneity = 0.14) and 1.010 (95% CI: 1.001; 1.019, six studies I^2^ = 0.0, *p* for heterogeneity = 0.49) for studies using long-term residential history. In terms of geographical region, the estimated RR was slightly higher in European compared to North American studies, with a pooled RR of 1.043 (95% CI: 1.017; 1.069, I^2^ = 6.6, *p* for heterogeneity = 0.37) for the three European studies and a pooled RR of 1.007 (95% CI: 0.999, 1.016, I^2^ = 0.0, *p* for heterogeneity = 0.90) for the seven North American studies.

The analysis by menopausal status showed a pooled RR of 1.014 for postmenopausal women (95% CI: 0.995; 1.033, nine studies, I^2^ = 0.0, *p* for heterogeneity = 0.51) and a pooled RR of 1.022 for premenopausal women (95% CI: 0.966; 1.085, six studies, I^2^ = 37.8, *p* for heterogeneity = 0.15).

The pooled analyses by hormone receptor status of the five studies showed a similar association as overall for either ER and PR positive breast cancer (pooled RR = 1.034, 95% CI: 0.992; 1.077, I^2^ = 0.0, *p* for heterogeneity = 0.52) or ER and PR negative breast cancer (pooled RR = 0.988, 95% CI: 0.925; 1.055, I^2^ = 6.6, *p* for heterogeneity = 0.37).

The sensitivity analysis showed that the results did not change when removing the study of Bai et al., which did not adjust for known breast cancer risk factors (i.e., variables related to lifestyle and reproductive factors) and which had the highest weight in the pooled effect [23] (Appendix A). Furthermore, excluding one study at a time did not influence the overall meta-estimate (Appendix A). In the further sensitivity analysis of postmenopausal women, the pooled estimate became significant adding two studies with over 80% of women at menopausal age (≥50 years old) [34,35] (pooled RR = 1.023, 95% CI: 1.004; 1.044). The cumulative meta-analysis showed that the RR was almost stable over time for breast cancer (Appendix A). When considering the ESCAPE project [26], using the included studies individually, the meta-estimate was very similar (pooled RR = 1.011, 95% CI: 1.003; 1.019), as well as the heterogeneity across studies (I^2^ = 0.0, *p* for heterogeneity = 0.50) (Appendix A). The meta-estimate did not change substantially when restricting to the 5 out of the 13 studies rated as being of high quality (NOS score ≥ 8) [25,35,48,49,50] (pooled RR = 1.03, 95% CI: 1.00; 1.06). 

Regarding NO_x_ exposure, three studies (all cohort studies) were included in the meta-analysis. Figure 4 shows the study-specific and pooled HRs and 95% CIs of invasive breast cancer for an increase of 10 µg/m^3^ of NO_x_ exposure. The pooled estimate was 1.023 (95% CI: 0.992; 1.054), with a significant heterogeneity (I^2^ = 75%, *p* = 0.02).

## 4. Discussion

We performed a systematic review of the literature on the association between breast cancer risk and main indicators of exposure to TRAP (namely concentrations of NO_2_, NO_x_, and BaP, distance to or length of major roadways, and traffic density). The results of the meta-analysis on NO_2_ (four case–control, one nested case–control, and eight cohort studies) indicated an effect of a 10 µg/m^3^ increase in NO_2_ exposure as a proxy for TRAP exposure on breast cancer risk with low heterogeneity between studies, although the magnitude of this effect was low. No association was found for NO_x_. Individual studies investigating BaP exposure or spatial surrogate variables of TRAP provided little evidence on a relationship with breast cancer risk.

The 21 studies reviewed including 7 case–control, 1 nested case–control, and 13 cohort studies were primarily from North America or Western Europe and varied in terms of the exposure assessment methods and the exposure time points and periods considered. While all studies on NO_2_ reported high mean concentration estimates, above the level of current WHO air quality guidelines (10 µg/m^3^) [54], the meta-analyses of NO_2_ stratified by continent suggest a possible variability between geographical areas with slightly higher risk estimates in Europe compared to North America. This is potentially in line with observations of varying exposures between European and North American cities [55]. However, the confidence intervals in the two regions were overlapping. Subgroup meta-analyses used to investigate heterogeneous outcomes showed similar risk estimates as overall, for cohort studies, studies using LUR models for NO_2_ exposure assessment, for postmenopausal women, and in strata of ER/PR status of the breast tumor. 

Our findings are in line with hypotheses from ecological studies showing correlations between trends in NO_x_ emissions or other traffic-related pollutants over time and breast cancer incidence [56,57,58]. Furthermore, the magnitude of the risk estimate is comparable to those reported on exposure to NO_2_ and lung cancer risk (HR = 1.04, 95% CI: 1.01; 1.08) [7], as well as on all-cause mortality (HR = 1.02, 95 % CI: 1.01; 1.05) [8]. Moreover, after updating the previous meta-analysis by Gabet et al. (11 studies plus the 15 studies of the ESCAPE consortium) by including 2 recent studies [37,48], we obtained similar results. NO_2_ is a major component of TRAP, and the NO_2_ distance decay pattern, decreasing concentrations to the background level within 100 to 200 m, has been consistently correlated with increasing distance from a road [59]. Furthermore, the concentration gradients for most traffic pollutants, i.e., VOCs and various particle species, have been shown to be well correlated with those for NO_2_ [56]. Therefore, NO_2_ has been proposed as a reasonable surrogate for assessing the contribution of traffic emissions to ambient air pollution and for estimating traffic exposure in epidemiological studies. Yet, while traffic emissions can contribute to up to 80% of ambient NO_2_ in cities, NO_2_ is not unique to emissions from motor vehicles, and confounding from other sources cannot be ruled out. Of note, most of the included studies used LUR models adjusted on measurements in ambient air [54] to estimate NO_2_/NO_x_ exposure [23,26,29,34,35,41,45,48,49,50]. LUR models, widely used in epidemiological studies, appear to be very appropriate for the assessment of NO_2_ exposure, and these models are relatively robust in capturing the small area variation of NO_2_ in comparison to the other models. Yet, the three studies using several methods to assess NO_2_ exposure yielded consistent results across the different methods [35,48,50]. Conversely, a recent meta-analysis on breast cancer risk and exposure to PM_2.5_ and PM_10_, both known to be correlated with NO_2_ exposure [56], did not show a significant association [17]. However, PM is complex because it is characterized by both the size and the chemical composition and includes various types of chemicals that are not only emitted from traffic [51,57].

Confidence in evidence from observational studies is enhanced if it is coherent with biological knowledge, including information about the latency and timing of the exposure. The biological mechanisms of the effect of TRAP on breast cancer risk are not fully understood. NO_2_ could act through endocrine disrupting and carcinogenic properties [15,58,60]. In addition, nitric oxide (NO), one of the two major nitrogen oxides associated with combustion sources, which oxidizes in air and forms NO_2_, has been reported to play a role in several stages of cancer, including angiogenesis, apoptosis, cell cycle, invasion, and metastasis [61,62,63]. NO can directly inhibit caspase activity, which is an effective way to block apoptosis [64,65,66], and can promote breast cancer development through the estrogen and progesterone pathways, both of which are involved in breast cancer carcinogenesis [62,67]. The observed positive results may also be explained by the surrogate role of NO_2_ for other pollutants present in traffic exhaust, including NMVOCs, which have been investigated and related to the development of mammary tumors in mice [68]. PAHs originating from traffic were linked to breast cancer with a strong association in women with certain biologically plausible DNA repair genotypes [13]. Moreover, PAHs may have an effect on oxidative stress, adding to the estrogenic and antiestrogenic [15] and methylation [14] effects on breast carcinogenesis, as well as DNA damage via the formation of adducts [13]. It has also been suggested that air pollution may increase breast cancer risk by increasing breast density [69], a strong risk factor for breast cancer [70]. However, results on studies on air pollution exposure and breast density have not to date been conclusive [69,71]. A recent systematic review on environmental exposures and breast cancer risk in the context of susceptibility (i.e., breast cancer family history; early onset breast cancer; and/or genetic susceptibility) reported that 74% of publications (20 out of 27 publications) found a statistically significant direct association with environmental exposures including PAHs and TRAP, in women with greater genetic susceptibility [72]. This is due to variants in carcinogen metabolism, DNA repair, oxidative stress response, cellular apoptosis, and tumor suppressor genes [72]. A recent review on epigenetic responses to TRAP exposure reported that there is no evidence in favor of a mediation of the association between NO_2_/NO_x_ exposure and increased breast cancer risk through epigenetic modifications. Yet, the authors suggested that PAHs and NO_2_ exposures may alter the methylation of breast tumorigenic genes (e.g., *EPHB2*, *LONP1*) [67].

This meta-analysis presents several strengths. A large number of studies representing a total of nearly 120,000 breast cancer cases were included, allowing sufficient statistical power to detect significant associations and to perform subgroup analyses. Although one study accounted for more than 50% of the analysis, its exclusion did not change the results for NO_2_. Heterogeneity was low for the NO_2_ analysis, but high for the NO_x_ analysis; however, we performed a random-effects model that controlled for heterogeneity. Individually, the included studies did not show a statistically significant association, suggesting a low potential publication bias, supported by the non-significant Egger’s test. Another strength of our study is the use of continuous exposure, allowing more accurate estimates than exposure in categories. Our meta-analysis mainly included prospective cohort studies, which are generally less prone to bias than case–control studies. The large majority of studies adjusted for the most-relevant confounders, i.e., age, reproductive and hormonal factors (age at menarche, oral contraceptive use, parity, breastfeeding, and menopausal status), body mass index, tobacco smoking, alcohol consumption, physical activity, education level or social class, hormone replacement therapy use, and family history of breast cancer. Bai et al. did not perform a fully adjusted model (the authors adjusted for age and socio-geographical data related to the women’s address) [23]; however, sensitivity analysis excluding this study from the meta-analysis on NO_2_ did not change the meta-estimate. Finally, efforts were made to objectively evaluate the quality of the included studies despite the lack of a single obvious tool for assessing the quality of observational epidemiological studies. The Newcastle-Ottawa Scale used in this study has previously been criticized for potential inter-operator variability [73]. Independent evaluation by two investigators combined with arbitrage in case of disagreement was performed to increase reliability of scoring and respond to these limitations. 

Among the limitations of our meta-analysis, we can point out the heterogeneity of the time periods covered by the exposure estimates across studies, as well as the lack of temporal concordance with the periods during which the exposure is relevant to breast cancer in some studies. Some studies used the subjects’ address at baseline, while others used their residential history. Similarly, some studies estimated exposure at a point in time, while others estimated exposure over a longer time period. Furthermore, sometimes, the baseline address was used throughout the follow-up, not taking into account the potential relocation of subjects [29,44]. In another study, the temporal sequencing was not respected because the exposure was measured after diagnosis and at the baseline address (i.e., 20 years earlier) [41]. However, when excluding this study from the meta-analysis, the meta-analytical estimate did not materially change. This heterogeneity of considering time-varying exposure may limit the interpretation of the combined results. While many studies had a long follow-up, several studies examined exposures shortly before diagnosis, ignoring particular periods of breast cancer susceptibility such as prenatal development, puberty, and pregnancy. During these specific windows, breast tissue undergoes structural and functional changes, as well as changes in the microenvironment and hormone signaling [15,74]. Nie et al. considered this in the WEB study and interestingly reported an increased risk of breast cancer among premenopausal women highly exposed to traffic emissions at menarche and among postmenopausal women exposed at first birth [47].

The distance of residence to traffic or a roadway can be a marker of socioeconomic status and may thus confound the association between breast cancer and TRAP exposure [75,76]. Of note, most studies adjusted for education level, social class, or socio-economic status. Moreover, exposure assessment in all studies was limited to exposure at the residential address. Yet, exposure at the workplace, as well as during commutes contributes to overall TRAP exposure, the latter, due to the proximity of emitting sources and high emissions during peak hours, being responsible for up to 21% of the daily air pollution exposure and 30% of the total daily dose of air pollutants inhaled [77] because of the proximity of emitting sources and high emissions at peak hours [78]. In addition, further studies are needed to evaluate simultaneous exposure to multiple traffic-related air pollutants and their potential additive or synergic effect.

## 5. Conclusions

Since traffic is the major contributor to NO_2_ concentrations and is responsible for high concentrations measured near busy roads, our findings suggest that TRAP is likely to increase breast cancer risk and provide additional support for the carcinogenicity of air pollution. However, because of the small magnitude of the effect (i.e., 1.5% increased risk for each 10 µg/m^3^ increase in NO_2_ exposure) and the heterogeneity of studies using TRAP exposure surrogate variables other than NO_2_, the certainty of evidence for an association between TRAP exposure and breast cancer risk remains moderate. Yet, our results contribute useful information to decision-makers and prevention policies, given that TRAP exposure affects a vast proportion of the population and, thus, potentially substantial consequences at a population scale. However, further studies are needed to confirm the current conclusions, in particular by using the finest possible exposure assessment and by considering exposure as a whole and at specific critical periods of susceptibility for breast cancer, as well as the effect of the pollutants as a mixture.

## Figures and Tables

**Figure 1 cancers-15-00927-f001:**
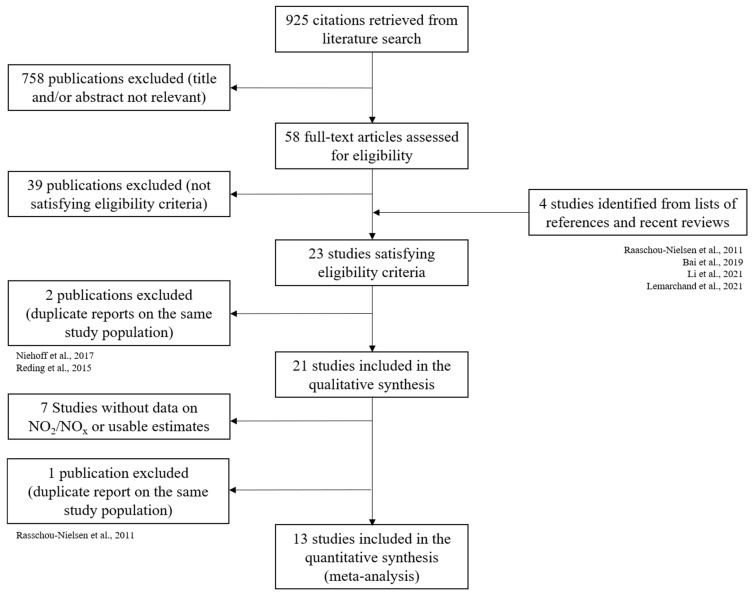
Flow chart of the selection process of articles included in the systematic review of the literature on breast cancer risk and exposure to traffic-related pollution and in the meta-analysis of observational studies on breast cancer risk and exposure to NO_2_ and NO_x_ [22,23,24,25,36,38].

**Figure 3 cancers-15-00927-f003:**
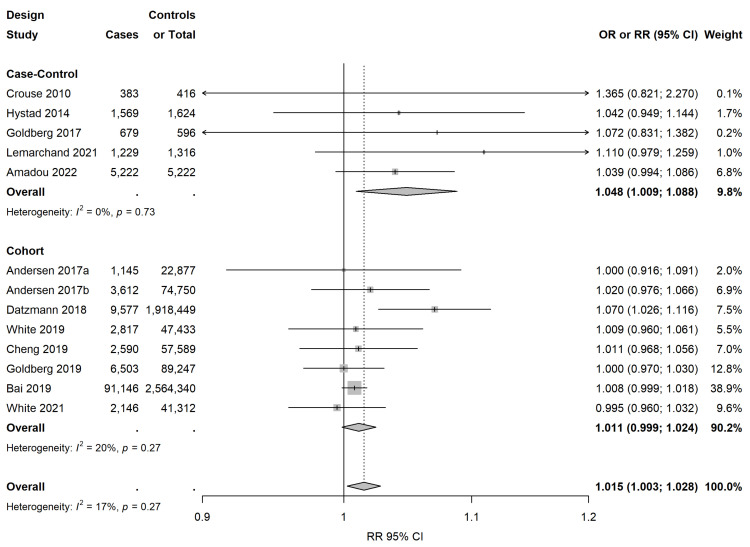
Study-specific and summary RRs and 95% confidence intervals for the association between invasive breast cancer and exposure to an increase of 10 µg/m^3^ in NO_2_ [23,25,26,29,34,35,39,40,41,45,48,49,50].

**Figure 4 cancers-15-00927-f004:**
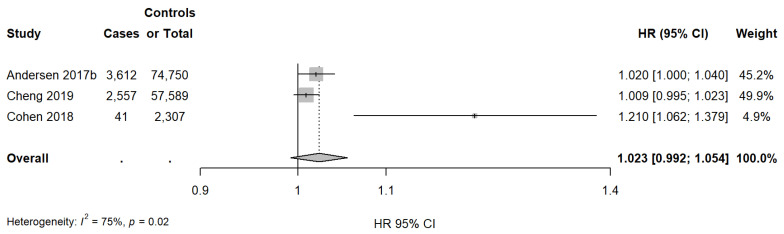
Study-specific and summary HRs and 95% confidence intervals for the association between invasive breast cancer and exposure to an increase of 10 µg/m^3^ in NO_x_ [26,35,44].

**Table 1 cancers-15-00927-t001:** Description of the twenty-one studies on traffic-related air pollutant exposure and breast cancer risk included in the systematic review.

Reference; Region	Study ID ^a^	Exposure	Design	No. of Cases/No. of Total Subjects or No. of Controls ^b^	Exposure Assessment	Variables of Adjustment	Quality Assessment (NOS Score/9)	Important Findings
Lewis-Michl et al., 1996; USA [46]	-	Traffic density (number of vehicle-miles divided by the number of miles of highway)	CC	793/966	Traffic density = vehicle miles/miles of highway (5-km^2^ grid cells) Weighted annual average of residential address exposures 20 years prior to the index date → based on residential history High density = average ≥ 100,000 vehicle miles/miles of highway	Age, family history, history of benign breast disease, age at first birth, education	4	High-density traffic (Nassau): OR = 1.29 (0.77, 2.15) High-density traffic (Suffolk): OR = 0.89 (0.40, 1.99)
Nie et al., 2007; USA [47]	-	BaP	CC	1068/1944	Exposure to BaP estimated by a geographical model validated and calibrated with measurements at different points in time of life → based on residential history	Age; ethnicity; education; BMI; smoking history; diet; medical history; age at first birth, number of births; family history; age at menarche; age at menopause; history of benign breast disease	8	At menarche Premenopausal: Q4/Q1: OR = 2.07 (0.91, 4.72) At first birth Premenopausal: Q4/Q1: OR = 1.22 (0.44, 3.36) Postmenopausal: Q4/Q1: OR = 2.58 (1.15, 5.83) 20 years prior to diagnosis Premenopausal: Q4/Q1: OR = 1.29 (0.59, 2.82) Postmenopausal: Q4/Q1: OR = 0.82 (0.58, 1.18) 10 years prior to diagnosis Premenopausal: Q4/Q1: OR = 1.49 (0.65, 3.43) Postmenopausal: Q4/Q1: OR = 0.80 (0.55, 1.17)
Crouse et al., 2010; Canada [29]	1	NO_2_	CC	383/416	Statistical methods (LUR) Annual average in 1985 (10 years before diagnosis), 1996 (at diagnosis), and 2006 at the interview address (1996–1997) ^c^	Age at diagnosis; age at menarche; age at first birth; duration of breastfeeding; age at bilateral oophorectomy; BMI; smoking status; alcohol consumption; education; hospital of diagnosis; family history; ethnicity; oral contraceptive use; duration of HRT; respondent status; history of benign breast disease; occupational exposure to solvents, low magnetic fields, CO, and PAHs; neighborhood income and SES	6	Postmenopausal: (per 5 ppb) 2016: OR = 1.52 (0.82, 2.81) 1996: OR = 1.42 (0.81, 2.48) Mean of 1996 and 1985: OR = 1.34 (0.83, 2.16) 1985: OR = 1.28 (0.84, 1.93)
Raaschou-Nielsen et al., 2011; Denmark [22]	-	Traffic density (distance from subject’s residence to the closest major road; number of vehicles on the nearest road)	Co	987/27,735	-Presence of a street with a traffic density >10,000 vehicles per day within 50 m of the residence -Total number of kilometers travelled by vehicles within 200 m of the residence each day → At baseline address	Age; smoking status; smoking intensity and duration; second-hand smoking; physical activity; BMI; diet; alcohol consumption; breastfeeding; number of births; age at first full-term pregnancy; previous benign breast tumor; previous diagnosis of hypertension; oral contraceptive use; HRT use; skin sensitivity to the sun	8	Major street within 50 m (yes/no): IRR = 0.98 (0.78, 1.22) Per 104 vehicle km/day within 200 m: IRR = 0.98 (0.88, 1.10)
Hystad et al., 2015; Canada [48]	2	NO_2_/traffic density (distance from subject’s residence to the closest major road)	CC	1569/1624	Dispersion model (CTM based on satellite data), statistical methods (interpolation and LUR) Average exposure (1975–1994) from postal codes of residential history Number of years participants resided within 50 m, 100 m, and 300 m of a highway or main road during the 20-year exposure period (1975–1994) → based on residential history postal codes	Age; study province; age at menarche; years of menstruation; parity; age at first birth; breastfeeding; oophorectomy; BMI; smoking status; years since smoking cessation; alcohol consumption; median household income; years of education; second-hand smoking status; meat and vegetable consumption; physical activity; mammography; neighborhood SES; time in urban area	8	NO_2_: OR = 1.04 (0.95; 1.14) Highways (for an increment of 1 additional year of residence) Years ≤ 50 m OR = 0.95 (0.73, 1.23) Years ≤ 100 m OR = 0.95 (0.78, 1.15) Years ≤ 300 m OR = 0.98 (0.86, 1.11) Major roads (for an increment of 1 additional year of residence) Years ≤ 50 m OR = 0.94 (0.82, 1.08) Years ≤ 100 m OR = 1.03 (0.92, 1.15) Years ≤ 300 m OR = 0.98 (0.90, 1.08)
Mordukhovich et al., 2016; USA [37]	-	PAH	CC	1274/1334	Geographical model validated and calibrated with measurements → based on residential histories in Nassau and Suffolk counties only; addresses at which a woman resided for at least 1 year	Age; education; ethnicity; religion; parity; BMI; age at first full-term pregnancy; oral contraceptive use; alcohol consumption; physical activity; breastfeeding; HRT use; SES	8	Traffic PAH exposure: - in 1995: 95th/<50th: OR = 1.06 (0.70, 1.60) - in 1960–1990: 95th/<50th: OR = 1.47 (0.70, 3.08)
Hart et al., 2016; USA [42]	-	Traffic density (distance from subject’s residence to the closest major road)	Co	3072/111,545	Distance of 3 types of roads to the address (time-dependent variable) using the 2007 roads database → based on residential history (every 2 years) (2007 roads database)	Age; ethnicity; family history; age at menarche; parity; age at first full-term pregnancy; height; BMI; BMI at age 18; history of benign breast disease; alcohol consumption; diet; oral contraceptive use; menopausal status; hormone use; smoking status; physical activity; individual SES; neighborhood SES	7	0–49 m from A1–A3 roads (vs. ≥ 200 m): HR = 1.02 (0.75, 1.37) 0–49 m from A1–A2 roads (vs. ≥ 200 m): HR = 1.44 (0.71, 2.92) 0–199 m from A1 roads (vs. ≥ 200 m): HR = 1.52 (0.89, 2.60)
Shmuel et al., 2017; USA [43]	-	Traffic density (characteristics of the main road and of the nearest cross-street)	Co	2028/42,934	Distance to the nearest intersection/cross-street Characteristics of the main road (number of lanes, presence of median or barrier) Traffic volume during rush hour → At the longest residence before 14 years old	Age, ethnicity; childhood SES; smoking status; education, family history; menopausal status; childhood residence urban/rural status	5	Characteristics of the main road at childhood residence: ≥3 lanes (vs. 1–2): HR = 0.8 (0.6, 1.1) With median or barrier of any kind (vs. without) HR = 1.2 (0.9, 1.7) Heavy traffic (vs. light traffic): HR = 0.9 (0.7, 1.1) Characteristics of the nearest cross-street or intersecting road: Within 100 ft., 3+ lanes and/or median/barrier and heavy traffic (vs.100 ft. + and/or (neither 3+ lanes, nor median/barrier)) HR = 1.4 (1.0, 1.9)
Goldberg et al., 2017; Canada [49]	3	NO_2_	CC	679/596	Statistical methods (LUR) Annual average in 2005–2006 at baseline address	Age; age at menarche; parity; age at first birth; breastfeeding; oophorectomy; BMI; smoking status; alcohol consumption; education; family history; ethnicity; oral contraceptive use; HRT use; second-hand smoking; marital status; census variables (immigrants; unemployment; education; income)	9	Postmenopausal: per increase of IQR 3.75 ppb: OR: 1.07 (0.83; 1.38)
Andersen et al., 2017a; Denmark [40]	4	NO_2_	Co (Danish Nurse Cohort Study)	1145/22,877	Method combining a dispersion model (THOR) and a proxy using GIS (Danish AirGis) 3-year annual running average from 1990 to index date based on residential history	Age; age at menarche; parity; age at first birth; BMI; smoking status; alcohol consumption; physical activity; menopausal status; oral contraceptive use; HRT use; urbanization level	7	NO_2_: HR = 1.00 (0.94–1.07) (per interquartile range increase 7.4 μg/m^3^)
Andersen et al., 2017b; 15 cohorts from nine European countries ^d^ [26]	5	NO_2_/NO_x_/traffic Density (number of vehicles on the nearest road)	Co (ESCAPE)	3612/74,750	Statistical methods (LUR) At the address at baseline Number of vehicles per day on the nearest road → At baseline address	Age; parity; age at first birth; breastfeeding; BMI; smoking status; years since smoking cessation; alcohol consumption; education; employment; physical activity; oral contraceptive use; HRT use; neighborhood income	7	Postmenopausal NO_2_: HR = 1.02 (0.98–1.07) *p* = 0.33 (per increase of 10 μg/m^3^) NOx: HR = 1.04 (1.00–1.08) *p* = 0.04 (per increase of 20 μg/m^3^)
Datzmann et al., 2018; Germany [34]	6	NO_2_	Co	9577/1,918,449	Statistical methods (LUR) Annual average in 2007 in the residential district at the baseline address	Age; alcohol-related disorder; unemployment; district number of physician contacts, population density and proportion of unemployment	6	NO_2_: RR = 1.07 (1.03–1.12) (per increase of 10 μg/m^3^)
Cohen et al., 2018; Israël [44]	7	NO_x_	Co	41/2307	Statistical methods (LUR) Average of estimated annual concentrations between 2004 and 2012 at baseline address	Age; smoking status; neighborhood SES; ethnicity; hypertension; diabetes; chronic heart failure; renal failure; hemoglobin levels	5	NOx: adjusted HR = 1.43 (1.12–1.83) (for a 10-ppb increase)
White et al., 2019; USA [39]	8	NO_2_	Co (The Sister Study)	2203/47,433	Statistical methods (kriging) Annual average in 2006 at baseline address	Age; parity; BMI; smoking status; education; ethnicity; HRT use; income; marital status; census tract level income; geographic region	7	All: HR = 1.06 (1.01–1.11) Invasive: HR = 1.01 (0.96–1.07) In situ (DCIS): HR = 1.23 (1.12–1.36) (for an increase in the IQR difference 5.8 ppb)
Goldberg et al., 2019; Canada [41]	9	NO_2_	Co (Canadian National Breast Screening Study)	6503/89,247	Statistical methods (LUR) Annual average in 2006 at baseline address	Age; age at menarche; pregnancy; BMI; smoking status; education; employment; occupation; family history; oral contraceptive use; HRT use; breast self-examination; contextual measures (education; income; unemployment)	6	Premenopausal: Rate ratio: 1.17 (1.00–1.38) for increase of 9.7 ppb Postmenopausal Rate ratio: 1.00 (0.95–1.06) for increase of 9.7 ppb
Bai et al., 2019; Canada [23]	10	NO_2_	Co (Ontario Population Health and Environment Cohort)	91,146/2,564,340	Statistical methods (LUR) Time-varying variables using a 3-year running average without taking into account the 4 years before the index date, using residential history	Age; census tract-level recent immigrants, unemployment rate, education and income; urban residency and a north/south indicator	7	NO_2:_ HR = 1.02 (0.99–1.04) for 8.2 ppb
Cheng et al., 2020; USA [35]	11	NO_2_/NO_x_	Co (The multi-ethnic cohort study)	≃2, 500 ^e^/57,589	Statistical methods (kriging, LUR), dispersion model A set of cumulative average exposures for a series of time intervals during monitoring from residential history	Age; age at menarche; parity; age at first birth; BMI; smoking status; alcohol consumption; family history; ethnicity; HRT use; menopausal status; physical activity; energy intake; neighborhood SES and education	8	NO_x_ Kriging: HR = 1.12 (0.96–1.31) (increase of 50.2 ppb) LUR: HR = 1.08 (0.96–1.22) (increase of 45.6 ppb) CALINE4: HR = 0.97 (0.73–1.26) (increase of 8.7 ppb) NO_2_ Kriging: HR = 1.09 (0.91–1.31) (increase of 16.5 ppb) LUR: HR = 1.04 (0.90–1.20) (increase of 18.6 ppb)
White et al., 2021; USA [45]	12	NO_2_	Co	2146/41,312	Statistical methods (LUR) Annual average in 2000–2010 at baseline residence and time-varying air pollution exposure throughout follow-up	Age; education; smoking status; parity; HRT use; BMI; census geographic region; menopausal status and menopausal status*BMI.	7	HR = 0.94 (0.87–1.02) for 9.90 ppb increase
Lemarchand et al., 2021; France [25]	13	NO_2_	CC (CECILE study)	1229/1316	Chemistry transport model 10-year period prior the reference date using residential history	Age; study area; family history; age at first full-term pregnancy; HRT use; physical activity	9	OR = 1.11 (0.96; 1.26)
Li et al., 2021; Taiwan [24]	-	NO_2_	Co	1603/98,017	Monitoring station	Age; monthly income and urbanization level	6	Q4/Q1 NO_2_: HR = 1.79 (1.48, 2.15)
Amadou et al., 2022; France [50]	14	NO_2_	Nested CC (XENAIR)	5222/5222	Statistical methods (LUR), chemistry transport model Cumulative exposure from inclusion to index date at each address from inclusion to index date	Age; date; department of residence; menopausal at baseline; physical activity; smoking status; education; rural urban status at inclusion; BMI; family history; history of benign breast disease; age at menarche; parity and age at first full-term pregnancy; breastfeeding; oral contraceptive use and HRT use	8	RR = 1.04 (0.99; 1.09)

BaP: benzo[a]pyrene, BMI: body mass index, CC: case–control study, Co: cohort study, CO: carbon monoxide, HRT: hormone replacement therapy, LUR: land use regression, PAH: polycyclic aromatic hydrocarbon, PM: postmenopausal women, SES: socio-economic status. ^a^ Used to indicate studies that contributed to the subgroup analyses in Table 2 and Appendix A. ^b^ Total of subjects for cohort studies or controls for case–control studies. ^c^ In the meta-analysis, the OR considered was estimated from the mean of 1985 and 1996. ^d^ Austria (n = 1), Denmark (n = 1), France (n = 1), Italy (n = 2), Netherlands (n = 2), Norway (n = 1), Spain (n = 1), Sweden (n = 5), the United Kingdom (n = 1). ^e^ Kriging: n = 2693 (NO_x_), n = 2727 (NO_2_); LUR: n = 2557 (NO_x_), n = 2590 (NO_2_); CALINE4: 2352 (NO_x_).

**Table 2 cancers-15-00927-t002:** Summary relative risks (RRs) and 95% confidence interval (CI) of breast cancer for an increase of 10 µg/m^3^ of NO_2_, overall and by strata of selected covariates.

		N Studies	RR (95% CI)	I^2^ (%)	*p* for Heterogeneity	*p* for Heterogeneity between Strata	ID of Included Articles ^a^
Overall meta-estimate		13	1.015 (1.003; 1.028)	16.9	0.27		1–6, 8–14
Exposure assessment method						1.00	
	LUR	10	1.016 (1.002; 1.030)	26.5	0.20		1–3, 5, 6, 9–12, 14
	Other	6	1.037 (1.006; 1.069)	9.8	0.35		2, 4, 8, 11, 13, 14
Address used for exposure assessment						0.24	
	At baseline	7	1.018 (0.993; 1.043)	38.7	0.14		1, 3, 5, 6, 8, 9, 12
	Residential history	6	1.010 (1.001; 1.019)	0.0	0.49		2, 4, 10, 11, 13, 14
Geographic area						0.10	
	North America	8	1.007 (0.999; 1.016)	0.0	0.90		1–3, 8–11, 12
	Europe	5	1.043 (1.017; 1.069)	6.6	0.37		4–6, 13, 14
Menopausal status						0.76	
	Postmenopausal	9	1.014 (0.995; 1.033)	0	0.51		1–3, 5, 8, 9
	Premenopausal	6	1.022 (0.963; 1.085)	37.8	0.15		2, 8, 9
ER/PR status of the tumor						0.32	
	ER+PR+	5	1.034 (0.992; 1.077)	0.0	0.52		3, 8, 11, 13, 14
	ER−PR−	5	0.988 (0.925; 1.055)	6.6	0.37		3, 8, 11, 13, 14

Note: All estimates are from random-effects models, and heterogeneity was evaluated through a Chi-squared test and quantified through the I^2^ statistic. LUR: land use regression, ER: estrogen receptor of the tumor, PR: progestative receptor of the tumor. ^a^ Study IDs are listed in Table 1.

## Data Availability

The identified data are available upon request to the corresponding author.

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
