# Peer review of "Traffic-Related Air Pollution and Breast Cancer Risk: A Systematic Review and Meta-Analysis of Observational Studies"

_cancers, 2023, doi:10.3390/cancers15030927_

Round 1
Reviewer 1 Report
The manuscript entitled “Traffic-related air pollution and breast cancer risk: a systematic review and meta-analysis of observational studies” presents a comprehensive review of the literature on air pollution exposure from traffic and breast cancer. The literature review and meta-analysis methods are well-described and the results appear to be robust given the number of sensitivity analyses and other stratified analyses presented with consistent results throughout. I have included a few minor comments below for the authors to consider.
- In the “Simple Summary,” it is suggested to maybe clarify the statement in Lines 16-18 (“Temporal concordance between the exposure periods relevant to breast cancer and the time period of the exposure assessment needs to be improved.” It is not clear who or what study this is directed to? Each study is different and has its own set of objectives and restrictions in exposure assessment. Perhaps replace with something more observational and stating that there is variability in exposure assessment approaches and length of follow-up time across studies and leaving out the “needs to be improved.”
- There is a lot of emphasis on the exposure assessment methods across the studies (which completely makes sense), but I didn’t see any description on how breast cancer cases were captured across the studies (e.g., identified from cancer registries)?
- It is suggested to reduce language in the text related to “not significant” or “significant” and replaced with language simply describing the effect estimates and perhaps describe the precision of the 95% confidence intervals.
o Kenneth J. Rothman and Sander Greenland, 2005: Causation and Causal Inference in Epidemiology. Amer J of Public Health. https://doi.org/10.2105/AJPH.2004.059204
o Greenland S, Senn SJ, Rothman KJ, Carlin JB, Poole C, Goodman SN, Altman DG. Statistical tests, P values, confidence intervals, and power: a guide to misinterpretations. Eur J Epidemiol. 2016 Apr;31(4):337-50. doi: 10.1007/s10654-016-0149-3.
- Related to the previous comment, I found it interesting in the description of results by ER/PR status (on lines 358-360) says “no association” was observed although the effect estimate is very similar to all other effect estimates in the sensitivity analyses presented in Table 2 as well as the results of the primary meta-analysis (which is described as “a significantly increased risk”). My point is that all the pooled effect estimates in Table 2 are extremely similar (within ~0.04 across the estimates in Table 2), and I would avoid saying there is “no association” because the 95% CI crosses over 1.0.
- Was breast cancer risk assessed among only women in the study? I’m assuming yes especially given the analyses among premenopausal and menopausal, but it may be helpful to confirm that men with breast cancer were not included, if this is the case.
- Section 3.2 (line 228) says that studies published between 1996 and 2021 were identified; however, there is one paper (Amadou et al) that was published in 2022.
- Table 1 appears after Figure 2, but Table 1 is described in the text before Figure 2 – maybe consider re-arrangement to have Table 1 come before Figure 2.
- Figure 2 is a very nice figure and synthesizes a lot of information in a very effective way.
- The paper is well-written but would benefit from an editorial review. Some examples of things noted are below:
o Carefully check that all abbreviations are defined upon first use (for example, NMVOC, PM, BC)
o “Nitrogen dioxides” should be “nitrogen dioxide” (no ‘s’ on the end of “dioxide”)
o On line 305, should “unit” be “null value” (i.e., 1.0)?
o Table 1 spans several pages – have header row repeat on each page.
o On the Table 1 footnote, “benzo[a]pyren” is missing an ‘e’ at the end – should be “benzo[a]pyrene”
o Table 2, first column: Consider left justifying the row labels and then indenting the labels for the categories under each to make it easier to see which labels are the subheading and which are the categories under each subheading.
o Line 413: I think should be “decreasing in background concentrations” vs. “decreasing to back concentrations”?
o Line 446: Should “genotype” be plural?
Author Response
Thank you for your careful review and valuable comments helping to improve our manuscript. Please find below a point by point response to the comments.
1) Why the authors excluded studies regarding gene-environmental associations (lines 119-121)? Epigenetic modulation and altered gene expression can be the result of chronic exposure to air pollution and thus in carcinogenesis. (Environ Sci Pollut Res Int. 2021; 28(40): 55981–56002.) Why to exclude such observational studies?
The question whether the association of TRAP with breast cancer risk is mediated by epigenetic modifications or gene-environment interactions, is very pertinent. This question was beyond the focus of the present study. Also, a recent review by Sahay et al (doi: 10.2217/epi-2018-0158 ; Reference N°76 of our manuscript) investigated this question and reported that there are currently no data showing direct evidence for NO2/NOx exposure and increased BC risk mediated via epigenetic regulation. Yet, this review reported that polycyclic aromatic hydrocarbons (PAH) and nitrogen dioxide (NO2) exposures may alter methylation of breast tumorigenic genes (e.g., EPHB2, LONP1) (see lines 503-505 of our manuscript).
We have now modified this sentence as follows:
A recent review on epigenetic responses to TRAP exposure, reported that there is no evidence in favor of a mediation of the association between NO2/NOx exposure and increased breast cancer risk through epigenetic modifications. Yet, the authors suggested that
2) Would these results be similar regarding countries with not so strict measures for emission of air pollutants?
It is likely that not so strict measures for emission for TRAP emission may result in higher level of exposure and possible higher risk of breast cancer estimates. However, in our review, the countries in which the studies are conducted have similar restrictions in terms of air quality protection. The two studies with the higher mean concentrations (for example two studies included in the ESCAPE study, EPIC-Varese and EPIC-Turin with respectively mean of NO2 exposure of 44.2 and 53.0 µg/m3) showed a similar association with breast cancer (respectively RR=1.06 (0.97-1.15) and OR=1.05 (0.83-1.32)).
3) There are also some abbreviations (PM, PAHs) that needs to be addressed and some formatting of fonts used (e.g., line 423, 435, 451)
Thank you to have noticed these mistakes. We made the corrections as requested in the text.
Reviewer 2 Report
The authors present a systematic-review and meta-analysis of observational studies regarding TRAP and breast cancer risk. It is a very complicated issue since it deals with the long term exposure in air pollutants and their impact regarding a specific cancer type and not in general. The authors should be congratulated for their detailed work which is really interesting to read, very well organized and presented.
I have some comments that I would like to see a response regardless of the choice to include it in the text.
1) Why the authors excluded studies regarding gene-environmental associations (lines 119-121)? Epigenetic modulation and altered gene expression can be the result of chronic exposure to air pollution and thus in carcinogenesis. (Environ Sci Pollut Res Int. 2021; 28(40): 55981–56002.) Why to exclude such observational studies?
2) Would these results be similar regarding countries with not so strict measures for emission of air pollutants?
3) There are also some abbreviations (PM, PAHs) that needs to be addressed and some formatting of fonts used (e.g., line 423, 435, 451)
Author Response
Thank you for your careful review and valuable comments helping to improve our manuscript. Please find below a point by point response to the comments.
- In the “Simple Summary,” it is suggested to maybe clarify the statement in Lines 16-18 (“Temporal concordance between the exposure periods relevant to breast cancer and the time period of the exposure assessment needs to be improved.” It is not clear who or what study this is directed to? Each study is different and has its own set of objectives and restrictions in exposure assessment. Perhaps replace with something more observational and stating that there is variability in exposure assessment approaches and length of follow-up time across studies and leaving out the “needs to be improved.”
We reviewed the sentence in the simple summary as suggested (line 16).
- There is a lot of emphasis on the exposure assessment methods across the studies (which completely makes sense), but I didn’t see any description on how breast cancer cases were captured across the studies (e.g., identified from cancer registries)?
Thank you for this suggestion, we agree that this description is missing. We added the following sentence in the results part (line 243)
“The selection of cases was based on cancer registries in eight studies [48,23,28,44,26,25,37,45], hospital and physician list in six studies [31,39,46,47,49,50], on self-administered questionnaires in six studies [22,42,28,27,45,51] and on health insur-ance databases in two studies [36,41] (data not shown).”
- It is suggested to reduce language in the text related to “not significant” or “significant” and replaced with language simply describing the effect estimates and perhaps describe the precision of the 95% confidence intervals.
We agree with your point of view and carefully review our manuscript reducing the use of “significant” term.
o Kenneth J. Rothman and Sander Greenland, 2005: Causation and Causal Inference in Epidemiology. Amer J of Public Health. https://doi.org/10.2105/AJPH.2004.059204
o Greenland S, Senn SJ, Rothman KJ, Carlin JB, Poole C, Goodman SN, Altman DG. Statistical tests, P values, confidence intervals, and power: a guide to misinterpretations. Eur J Epidemiol. 2016 Apr;31(4):337-50. doi: 10.1007/s10654-016-0149-3.
- Related to the previous comment, I found it interesting in the description of results by ER/PR status (on lines 358-360) says “no association” was observed although the effect estimate is very similar to all other effect estimates in the sensitivity analyses presented in Table 2 as well as the results of the primary meta-analysis (which is described as “a significantly increased risk”). My point is that all the pooled effect estimates in Table 2 are extremely similar (within ~0.04 across the estimates in Table 2), and I would avoid saying there is “no association” because the 95% CI crosses over 1.0.
We agree with this point of view and modified the sentence (line 405) as follow:
“The pooled analyses by hormone receptor status of the five studies, showed a similar association as overall for either ER and PR positive breast cancer (pooled RR = 1.034, 95% CI: 0.992; 1.077, I²= 0.0, p for heterogeneity= 0.52) or ER and PR negative breast cancer (pooled RR = 0.988, 95% CI: 0.925; 1.055, I²= 6.6, p for heterogeneity = 0.37).”
- Was breast cancer risk assessed among only women in the study? I’m assuming yes especially given the analyses among premenopausal and menopausal, but it may be helpful to confirm that men with breast cancer were not included, if this is the case.
We selected only studies that addressed the risk of breast cancer in women. This was initially specified only enouncing the PECO strategy (line 91). We have now added a sentence in the “2.2. study selection” part (line 123) as follow:
“Studies on male breast cancers were not considered.”
- Section 3.2 (line 228) says that studies published between 1996 and 2021 were identified; however, there is one paper (Amadou et al) that was published in 2022.
We modified it in the text (line 230).
- Table 1 appears after Figure 2, but Table 1 is described in the text before Figure 2 – maybe consider re-arrangement to have Table 1 come before Figure 2.
We have shifted Table 1 before Figure 2.
- Figure 2 is a very nice figure and synthesizes a lot of information in a very effective way.
Thank you for this encouraging and very positive comment.
- The paper is well-written but would benefit from an editorial review. Some examples of things noted are below:
Thank you for all these corrections. We have carefully revised all of them accordingly in the text.
o Carefully check that all abbreviations are defined upon first use (for example, NMVOC, PM, BC)
o “Nitrogen dioxides” should be “nitrogen dioxide” (no ‘s’ on the end of “dioxide”)
o On line 305, should “unit” be “null value” (i.e., 1.0)?
o Table 1 spans several pages – have header row repeat on each page.
o On the Table 1 footnote, “benzo[a]pyren” is missing an ‘e’ at the end – should be “benzo[a]pyrene”
o Table 2, first column: Consider left justifying the row labels and then indenting the labels for the categories under each to make it easier to see which labels are the subheading and which are the categories under each subheading.
o Line 413: I think should be “decreasing in background concentrations” vs. “decreasing to back concentrations”?
o Line 446: Should “genotype” be plural?